# Refractive Properties of Conjugated Organic Materials Doped with Fullerenes and Other Carbon-Based Nano-Objects

**DOI:** 10.3390/polym15132819

**Published:** 2023-06-26

**Authors:** Natalia Kamanina

**Affiliations:** 1Vavilov State Optical Institute, Kadetskaya Liniya V.O. 5/2, 199053 St. Petersburg, Russia; nvkamanina@mail.ru; Tel.: +7-(812)-327-00-95; 2Department of Photonics, St. Petersburg Electrotechnical University (“LETI”), ul. Prof. Popova 5, 197376 St. Petersburg, Russia; 3Petersburg Nuclear Physics Institute, Part of Kurchatov National Research Center, 1 md. Orlova Roshcha, 188300 Gatchina, Russia

**Keywords:** modified nanostructured organics, sensitization, polyimide, fullerenes, carbon nanotubes, reduced graphene oxides, shungites, refractive and photoconductive features, surface

## Abstract

Due to the high demand for optoelectronics for use in new materials and processes, as well as the search for their modeling properties, the expansion of the functionality of modified materials using nanotechnology methods is relevant and timely. In the current paper, a specific nanotechnology approach is shown to increase the refractive and photoconductive parameters of the organic conjugated materials. The sensitization process, along with laser treatment, are presented in order to improve the basic physical–chemical properties of laser, solar energy, and general photonics materials. Effective nanoparticles, such as fullerenes, shungites, reduced graphene oxides, carbon nanotubes, etc., are used in order to obtain the bathochromic shift, increase the laser-induced change in the refractive index, and amplify the charge carrier mobility of the model matrix organics sensitized with these nanoparticles. The four-wave mixing technique is applied to test the main refractive characteristics of the studied materials. Volt–current measurements are used to estimate the increased charge carrier mobility. The areas of application for the modified nanostructured plastic matrixes are discussed and extended, while also taking into account the surface relief.

## 1. Introduction

It is well known that, at the beginning and middle of the last century, materials were fundamentally divided according to their electro-conducting properties, taking into account the charge carrier content, the changes in the dielectric constant (ε), and changes in the charge carrier’s mobility (μ) [1]. The preliminary breakdown of the materials according to the above-mentioned parameters is presented in Table 1.

However, with the extensive use of laser tools in the study of the characteristics of inorganic and organic materials, particularly the refractive properties of the features of nanostructured materials [2,3,4,5,6,7,8], which are undeniably connected to their spectral and photoconductive properties, these studies began to occupy a special research focus. Moreover, due to the effective use of the fullerenes and other nano-objects [9,10,11,12,13,14,15,16,17,18,19,20] in material parameter modifications, the sensitization (doping) process was established as the dominant method. Furthermore, this technique can be easily tested and visualized, permitting a good opportunity for effective analytical analysis. Thus, the photoinduced electron transfer in the conducting-polymer-C_60_ composites was evaluated using infrared photoexcitation spectroscopy [9]; the photoinduced electron transfer in the C_60_-doped poly(*N*-vinylcarbazole) films was revealed by picosecond laser photolysis in [12]; the significant effect of fullerene doping on the absorption edge shift was determined for the 2-cyclooctylamino-5-nitropyridine nanocomposite in [15]; refractive grating was recorded in the fullerene-doped liquid crystal materials in [5,11,16]; a high conducting polyaniline complex was synthesized with fullerene C_60_ in [14]; and the improvement of the photovoltaic properties of the solar cells, based on the poly(methyl phenyl silane)-C_60_ nanocomposites, was shown in [19].

It should be noted that the doping process of the different materials using the perspective nanostructures—fullerenes, carbon nanotubes (CNTs), quantum dots (QDs), graphene oxides (GrO), etc.—provokes the dramatic change in the basic physical–chemical material properties vital for the improvement of the technical device parameters [21,22,23,24,25,26,27,28,29,30,31,32,33]. Thus, the role of fullerene C60 in controlling the performance of inverted perovskite solar cells was shown in [21]. Some fullerene derivatives have been applied for solar energy and diode applications. Filter-free narrowband photomultiplication-type planar heterojunction (PHJ) organic photodetectors (PM-PHOPDs) were first realized by employing a thick front donor layer and an ultrathin PC71BM layer, as shown in [22]. Moreover, a smart strategy has been successfully proposed to improve the performance of PM-PPDs by inserting a poly-TPD layer as a buffer layer between the PEDOT:PSS/ITO and P3HT:PC61BM layers, in addition to decreasing the thickness of the P3HT:PC61BM layer; this technique was presented in [23,24].

The beneficial electrocatalytic properties of fullerenes and carbon nanotubes in sensor applications, which coincided with their electron affinity energy, were demonstrated in [25]. Multi-walled carbon nanotubes were investigated as potential mechanical reinforcement agents in two hosts, polyvinyl alcohol (PVA) and poly~9-vinyl carbazole (PVK), in [26]. It was found that, by adding various concentrations of nanotubes, both the Young’s modulus and the hardness increased by factors of 1.8 and 1.6 at 1 wt% in PVA, and 2.8 and 2.0 at 8 wt% in PVK. The multi-walled carbon-nanotube-reinforced phase, decorated by silver nanowires, was homogeneously dispersed in the polyamide acid solution during the in situ polymerization process, as shown in [27]. This method permits the formation of thermal transfer networks in this complex nanocomposite. The thermal conductivity of 0.4426 W/m × K was achieved with only a 3.0 wt% addition, while the glass transition temperature (T_g_) and heat-resistance index (THR) were increased to 295.65 °C and 297.88 °C, respectively. The polymer conductivity and local electrical field increase with the increase in graphene oxide content, as presented in [28], indicating an important significant decrease in the operating voltage. In [29], the role of graphene in provoking an increase in the surface area and the charge carriers in a polypyrrole-modified graphene sheet was established. An electrochemical supercapacitor has been fabricated using this layered nanocomposite as the electrode material to obtain a large specific capacitance value (~931 F/g). In the investigation in [30], the poly(2-ethyl-2-oxazoline)–polyvinylpyrrolidone–graphene nanocomposites were synthesized, and their properties were evaluated. The dielectric constant, dielectric loss, and tangent loss of the nanocomposites were decreased by increasing the frequency of the applied electric field, extending the application of this nanocomposite in electronic and optoelectronic devices. The minimum starting voltage of 1.91 V and the maximum brightness of 12,057 cd × m^−2^ were achieved in [31] in the light-emitting diodes based on the highly efficient green quantum dots CdSe/ZnS. A tunable liquid crystal core refractive index sensor based on surface plasmon resonance activated by a gold nanofilm was made in [32]. A good background review of the features of the nano-objects based on the fullerenes, carbon nanotubes and graphene was carried out in [33].

It should be mentioned that the nanostructuration process is now intensively considered for biomedicine applications as well. It was considered, for example, for tissue and bone tissue engineering [34,35]; for therapeutic nanoparticles-assisted living cell tracking [36], for cellular bio-imaging applications [37], etc. A good fundamental review of the advances in biophotonics in the 21st century was made in [38]. Moreover, the tendency to replace the nano-structuration doping with the bio-structuring one was established and explained in [39,40,41,42,43]. It was shown that bio-object-doped materials reveal the nonlinear optical coefficients close to those for the nano-object-sensitized ones that permit us to conclude the following main idea: the existence of an innovative tendency to use the non-toxic bio-objects instead the nano-object ones to optimize the organic photorefractivity [39,40]. The nonlinear susceptibility tuning in a functionalized DNA system was established in [41]; the rotation of the polarization plane of light via use of DNA-based structures for liquid crystal doping to apply in innovative medical devices was shown in [42]; and nano- and bio-particle-assisted optical bio-imaging was proposed in [43]. Some points of view of why the nano- and bio-object-doping dramatically influences the refractive properties, which greatly provokes the change of other materials’ physical characteristics, were discussed in [44].

Thus, as can be established by the dramatic change of the spectral parameters, conductivity, morphological, structural and refractive properties were interconnected with one another and can be more easily found via nanostructuration of the materials. In other words, via the nanotechnology approach, one can reveal the unique features of the novel composites. It should be noticed that in the nanoscale region there is a clear correlation between the spectral, structural, mechanical and refractive properties of the materials. This correlates with the optoelectronics request as a whole. Additionally, it should be mentioned that the rapid development of the optoelectronic devices, solar cells, and laser instruments poses an important task of creating and studying, from one side, the structures, which can be capable of effectively absorbing, limiting, converting, modulating, and recording the optical processing data in wide spectral and energy ranges; from another side, the new schemes and approaches can be capable of activating and simulating the modern features of the materials, not only in general photonics, but in biomedicine as well. Thus, the optoelectronics materials, especially doped ones, have the promising prospect to resolve most complicated technical tasks.

In this concept, it should be remarked once again that the indication, namely the change of the refractive properties, connected with the basic physical–chemical features, is very important. Alternatively, the doping process activates the creation of the intermolecular-interaction mechanism that influences the refractive and photoconductive parameters of the organic systems very intensively. In this case, the most important characteristic of the organic materials is the induced dipole, which can be expressed through dipole polarizability α^(n)^. [45] These are in turn related by the proportional dependence to the nonlinear susceptibility χ^(n)^ [46,47,48,49,50] and to the local volume υ of the materials. By way of example, we consider a small local volume of our medium that is substantially smaller than the incident wavelengths of 532, 633, 1064, and 1540 nm in the basic experiments; for comparison, the fullerene molecule dimensions are 0.65 ± 0.7 nm. Thus, laser–matter interaction provokes the change in the polarization of the media and predicts the change in the important properties, such as the dynamic, photorefractive and photoconductive ones.

In the current paper the large refractive index changes in the doped organic conjugated systems are discussed. The comparative results of the refractivity and charge carrier mobility changes are shown for the doped polyimides and pyridine materials sensitized with the fullerenes, shungites, and reduced graphene oxides. The improved refractive material parameters are tested via the laser technique and estimated by the classical formulas [47] applied for the nanostructured organics.

## 2. Materials and Methods

Fullerenes C_60_ and C_70_, reduced graphene oxides, and shungites were used as the effective nano-objects to sensitize the polyimide and 2-cyclooctylamino-5-nitropyridine (COANP) materials. C_70_ fullerenes have been purchased from Alfa Aesar (fullerene powder, 97%, type No.39720). The shungites (Sh) mixture has been received from Institute of Geology, Karelian Research Centre, Russian Academy of Sciences, (Petrozavodsk, Karelia, Russia). Reduced graphene oxides (RGO) have been received from Nanoinnova Technologies SL, Madrid, Spain (http://www.nanoinnova.com/). The content of the nanoparticles in the model plastic matrix was placed in the range of 0.1–5 wt.%. It should be remarked that the content of the nanoparticles coincided with the value of the electron affinity energy of the intramolecular acceptors. It means that the smaller the electron affinity was for an intramolecular acceptor, the greater the amount of the sensitizer (as an intermolecular acceptor) was introduced into the matrix. The 3–6.5% solutions in chloroform or in tetrachloroetane were prepared. It should be noted that tetrachloroetane had a very high solvent action for both the photosensitive organic matrixes and for fullerenes as well [51]. The films were developed using centrifuge deposition. The nanoparticle-doped film thickness was varied within 3–5 μm. The polymer films were placed onto the glass substrates covered with the transparent conducting layers based on ITO heterostructure. For the electric measurements, gold contacts were applied to the thin film’s upper side. The bias voltage applied to the photosensitive polymer layers has been varied from 0 to 70 V. The current–voltage characteristics have been measured for the samples with various concentrations of the nanoparticles additive under the conditions from dark to variable illumination intensity.

The basic photorefractive characteristics were studied using the four-wave mixing technique. Some variant of this scheme is shown in Figure 1.

The second harmonic of the nanosecond-pulsed Nd-laser (1) at the wavelength of 532 nm was used. Two beam splitters (3) are applied. The lens (11) placed after the sample (7) revealed the grating shown in the inset to the figure with the diffraction pattern itself. An additional Nd or He-Ne laser (6) is used to visualize the diffraction order when the reversible process can be transferred in the irreversible one. The energy density was chosen in the range of 0.1–0.9 J × cm^−2^. The amplitude-phase thin gratings were recorded under the Raman–Nath diffraction conditions at spatial frequency (Λ) of 90, 100 and 150 mm^−1^. Indeed, the condition for recording thin amplitude-phase holograms under the Raman–Nath conditions was implemented, when the thickness of the medium under the test is less than the period of the formed lattice. It should be remembered that photorefractive parameters were revealed not only for the nanosensitized films, but also for pure ones for comparison.

## 3. Results and Discussion

### 3.1. Change of the Refractive Parameters of the Body of Organics

It should be mentioned that the fullerenes C_60_, C_70_ and higher ones, as well as the shungites, etc., nanoparticles are basically successfully used to sensitize the organic conjugated materials with the initial donor–acceptor interaction process. In this case, the main reasons to use, for example, the fullerenes are connected with their high value of electron affinity energy and unique energy levels. The electron affinity energy of the fullerenes is ~2.65–2.7 eV, and that of the shungite structure is ~2 eV, which is larger than those for most dyes and organic molecules *intra*molecular acceptor fragments. It can stimulate the efficient *inter*molecular charge transfer complex (CTC) formation in the nano-objects-doped organic conjugated materials. Moreover, fullerenes can accept six or more electrons [52] that permit us to dramatically increase the delocalized charge. Furthermore, shungite also has some peculiarities [53,54], such as its own dipole moment, which is close to 2D. This evidence provokes us to make a novel nanocomposite with a larger dipole moment, in comparison with the analogous value for the pure organic matrix. A qualitative picture of the predominance of the charge transfer from an *intra*molecular donor, not to an *intra*molecular acceptor, but to an injected nano-object, is shown in Figure 2. Of course, it is worth noting that nanoparticles such as WS_2_ and CoFe_2_O_4_ are not carbon-containing, but they are taken into account in this model for the possibility of referring to it in subsequent experiments.

So many nano-objects can be incorporated in the organic matrix structure. These injected nanoparticles, as a rule, have a greater value of the electron affinity than an *intra*molecular matrix acceptor, have a branched surface, and often have their own dipole moment as well.

One can see that Figure 2 indicates the matrix organic with the initial *intra*molecular donor–acceptor interaction (Figure 2, the upper part) and the different types of the *inter*molecular acceptors (Figure 2, the lower part), which can be involved in the *inter*molecular CTC formation. Figure 2 shows the different ways for the CTC evidences via nanostructurization. Fullerenes, graphene oxides, shungites, carbon nanotubes, quantum dots, lanthanides nanoparticles (NPs), Janus NPs, WS_2_ nanotubes, CoFe_2_O_4_ NPs, as well as DNA can be incorporated into the organics in order to effectively modify their main features. In this case, the organic conjugated systems can be considered as perspective composites to obtain the advantage in the organic solar cells, optical limiters, modulators, switchers due to the extended spectral shift, increased charge carrier mobility, large refractive index and flexibility. It should be remarked that DNA bio-objects incorporated into the liquid crystal (LC)-conjugated materials can be considered in this model as well. It should be mentioned that most of the organic structures doped with the indicated nanoparticles in the additional experiments, such as mass-spectrometry analysis, color change testing, etc., have been made in order to support the *inter*molecular CTC formation.

Let us consider some of these nanoparticles features that influence the polymer matrix based on the polyimide in detail. Please note that the main monomeric links of the polyimides are *intra*molecular donor–acceptor (D–A) complexes with the electron charge transfer between the *intra*molecular donor and *intra*molecular acceptor fragments. Polyimides consist of acceptor diimide fragments with an electron affinity energy of 1.12–1.46 eV [55] and of the donor fragments as triphenylamine (TPA), carbazole, fluorene, and benzene with an ionization potential of 6.5–9.2 eV. Low ionization potential allowed the D–A complexes to be formed between monomeric links of the polyimide and incorporated molecules as the *inter*molecular acceptor during the sensitization process. Previously, it was shown in [55] that the dipole moment, dielectric constant and absorption cross-section were drastically increased after adding fullerene to the polyimide. In addition, for example, the color of the polyimide varies from bright orange (pure structure) to bright brown (when sensitized with the fullerene C_60_) or to dark orange (when sensitized with the fullerene C_70_).

Moreover, the 2-cyclooctylamino-5-nitropyridine (COANP) structures, doped with the same nanoparticles as the polyimide materials, revealed larger refractive parameters or the same ones but at larger energy densities. It coincided with the fact that the COANP *intra*molecular acceptor fragment is an NO_2_ group. This atomic group is bound to the donor fragment through the benzene ring. For a separate NO_2_ molecule or radical, the electron affinity is 2.3 eV, while the NO_2_ group bounded with the benzene ring has an electron affinity of only 0.54 eV [56]. Thus, it is smaller than the electron affinity of the fullerene by a factor of 4. It should be remarked that the sensitization of the COANP system with the 7,7,8,8,-tetracyanoquinodimethane dye was shown in [57]; the bathochromic spectral shift was established. After the fullerene-doping process application, the more substantial spectral red-shift was found [58].

Thus, the *inter*molecular charge transfer complex formation process permits us to find the drastic improvement of the spectral, photoconductive and dynamic parameters, to establish the change of the nonlinear optical characteristics of the nano-objects-sensitized organics compounds. Really, once again, an increase in the barrier-free path of the charge carrier transfer from an *intra*molecular donor to an *inter*molecular acceptor leads to an increase in the mobility of charge carriers; this stimulates an increase in the photoconductivity. An increase in the dipole moment is accompanied by an increase in the absorption cross-section (the square of the dipole moment is proportional to the absorption cross-section); which is realized in a change in absorption and spectral features. Thus, there is a correlation between the photoconductive and refractive parameters of sensitized materials.

Indeed, an important aspect should be mentioned. In order to analyze the nonlinear optical processes, one should take into account that, when the electric field of the electromagnetic wave is less than the intra-atomic electric field (the last one is correlated with the electron charge and with the Bohr radius), we should estimate the linear effect. But, when the electric field of the laser electromagnetic wave is larger than the intra-atomic electric field, we should draw attention to the nonlinear optical features. Based on this aspect, the values of the optical susceptibility play an important role in the nonlinear optical effect, as has been remarked in the introduction paragraph. Indeed, the optical susceptibility coincided with the change of the refractive index. The comparative data of the laser-induced refractive parameters changes are presented in Table 2 [49,59,60,61,62]. The previously published and current data are shown for the comparison. It should be noted that the refractive index change values for the fullerene-doped polyimide irradiated at the 1315 nm and the polymer-dispersed LC structure based on COANP + C_70_ treated at 532 nm were shown in [49,63] as well.

It should be mentioned that the laser-induced change of the refractive index Δ*n_i_* has been calculated via Equation (1). It is based on the mathematical procedure shown in [47] and applied for the nano-doped organics in [48,49,50] via measurement of the diffraction efficiency (η) at the Raman–Nath diffraction conditions. The realization of the Raman–Nath diffraction conditions has been checked at spatial frequency (Λ), when Λ^−1^ ≥ d, and shows the best energy spread by the diffraction orders.
(1)$$\eta =\frac{I_{1}}{I_{0}}={\left(\frac{\pi \Delta n_{i}d}{2\lambda }\right)}^{2}$$

Analyzing the data shown in Table 2, one can testify that in the case of a nanocomposite irradiated at small spatial frequencies (large periods of recorded grating), a drift mechanism of the carrier spreading in the electric field of an intense radiation field will probably predominate, while at large spatial frequencies (short periods of recorded grating), the diffusion process dominates.

Using the data shown in Table 2, one can calculate nonlinear refraction *n*_2_ and cubic nonlinearity χ^(3)^ of the composites studied. These parameters are placed in the range: *n*_2_ ~10^−10^–10^−9^ cm^2^ × W^−1^ and χ^(3)^ ~10^−9^–10^−8^ cm^3^ × erg^−1^.

Moreover, the data testify that the estimated Δ*n* in the current study for the thin-film doped organics (the sample thickness was 3–5 μm, Δ*n* ~10^−3^) is larger than that obtained, for example for the Si crystal [64] with the thickness of the sample of 50,220,470 μm and Δ*n* ~10^−5^. Furthermore, the nonlinear characteristics of the doped organic thin film materials studied in the current research in the visible spectral range are close to those for inorganic Si-based voluminous structures [65].

It should be remarked that the obtained nonlinearities for the thin-film doped organics can be the same or larger than the ones for classical inorganic crystal of LiNbO_3_. For example, the nonlinear refractive coefficient of the lithium niobate crystal estimated in [66] showed the nonlinear refractive coefficient value of *n*_2_ = 2.2 × 10^−9^ cm^2^ × W^−1^ for the terahertz range and was essentially smaller for the visible range.

Thus, it provokes the organic structures with nano-objects to be used in an extended area of optical and photoconductive applications.

It should be noticed that the evidence that the refractive properties increase coincided with the photoconductive features, which have been previously shown in publication [17,18] for the polyimide and COANP nanocomposites. This has been calculated via the mathematical procedure written in [1]. The charge carrier mobility in a fullerene-doped polyimide film has been estimated as ~0.3 × 10^−4^ cm^2^ × V^−1^ × s^−1^, while the carrier mobility of pure polyimide films has been tested as ~0.5 × 10^−5^ cm^2^ × V^−1^ × s^−1^. The obtained small carrier mobility of the pure polyimide materials was compared with that shown before in [67]. It should be noted that the increase in the photoconductive characteristics for the polyimide materials doped with the CNTs was previously shown in [68,69], which are also correlated with the photoconductive parameters estimated previously by us in [17,70].

Based on the analyzed results presented in Section 3.1, the following idea can be considered and supported by the illustrated data presented in [59,60,61]. It is possible that the elementary materials volume associated with the lattice parameters of the inorganic materials may correlate well with the local volume of the medium obtained by the irradiation of the organic matrices with the introduced nanostructures. Indeed, the interference pattern from two beams, recording the lattice, provides the diffraction due to the nanostructures incorporated, creating a 3D environment. The features of this 3D element coincide with the parameter characteristics of the entire volume of the organic system. Considering the diffraction pattern shown in [61,71] and presented in the inset of Figure 1, it can be seen that the spatial frequency (and hence the lattice period) along the *x* and *y* axes differs from the spatial frequency recorded along the *z* and *x* axes. This can provide devices with a denser recording of the optical information, which is in demand in general optoelectronics.

### 3.2. Surface Relief Influence on the Physical-Chemical Parameters of the Body of Organics

Returning to the current results, one can testify that the nanoparticles sensitization influences not only the basic material parameters, but also the surface relief. Firstly, this fact has been found especially for the polyimide materials doped with the different fullerene content [72]. The fullerene skeleton can efficiently influence the surface relief. Now, some similar results can be considered for the fullerene-doped COANP structure, which also indicated that the relief modification depended on the content of the nano-objects in the body of the matrix organic thin film. C_70_ fullerene has been used as the dopant in this case. It has been found that the wetting angle has been increased from 78–79 degrees up to 85–86 and 92–93 degrees when the content of the fullerenes has been increased as well from 0 wt.% to 0.5 wt.% and to 2.0 wt.%. It is connected with other features of the composites based on fullerene-doped COANP materials, for example, with the spectral parameter changes analyzed in [73]. It should be remarked once again that the effect of the dye and fullerene doping on the absorption edge shift in COANP has been shown in [57,58].

Based on the data shown in Table 3, one can see that the wetting angle for the water drops placed on the organic thin film surfaces is increasing via the content of the nano-objects increases. It was established for the polyimide, COANP, PMPS, PVA, NPP, and PNP structures [72,74]. Moreover, the first experiments were made to estimate the contact angle at the fullerene-doped [75] and WS_2_ nanotubes-doped polyimide [76] thin film surfaces when the LC drops were placed on the doped organics. The correlation between the concentration of injected nanoparticles and the surface relief of the organic matrices was discussed. It should be remarked that an interesting logical link can be postulated: for all the studied organic materials with an increase in the concentration of the nanosensitizers, introduced into the matrix organic base, both with the modification of surface roughness, and there is a shift in the transmission spectra towards long wavelengths.

Thus, the incorporated nanoparticles in the organic materials body, from one side, can form the novel relief, the properties of which are useful for the orientation of the LC molecules in the electrically- or light-addressed spatial light modulators or in display elements. From the other side, this relief can be used to explain the influence of an additional reflection and scattering on the optical limiting process.

It should be mentioned that the surface relief studied for the organic and inorganic surfaces, which influences different optoelectronic devices, was shown in [77] for the display applications; for efficiency of the diffractive optical elements used in the imaging optical systems [78]; for the control of the light scattering in [79], etc. Surface display [77] is a recombinant technology that expresses target proteins on cell membranes and can be applied to almost all types of biological entities, from viruses to mammalian cells. This technique was used for various biotechnical and biomedical applications such as drug screening, biocatalysts, library screening, quantitative assays, and biosensors. The study proposed in [78] showed the effect of surface roughness on the efficiency of the diffractive optical elements via application of the mathematical model, in which the basic object to study was the PMMA structure and a combination of PMMA and polycarbonate substrates operated in the visible spectral range. The results can be used to analyze the effects of surface roughness on the diffraction efficiency. The absorption loss under diffuse scattering is studied theoretically in [79] by applying a combination of the scattering matrix approach with diffraction theory for randomly nanotextured interfaces. It was shown that this modeling is in excellent agreement with the respective measurements.

It should be mentioned once again that some nanoparticles are provoked to form the relief as novel coatings [80,81,82,83,84], which are useful not only for general optoelectronics but for biomedicine and biotechnology. Classical optoelectronic materials and natural ones were considered as the substrates for the perspective coatings, which influence the basic material properties as well. It is important to note that the novel relief, as the coatings, can be also detected via application of the holographic scheme shown previously in detail in paper [85]. Moreover, other technique should be used in order to accept additional information about the structured materials. Focused electron beam induced processing allows finding important data on the structured novel coatings, when fullerene matrix used [86]. FTIR spectroscopy, X-ray diffraction, statistical analysis, etc. approaches can be applied to test the performance of the modified materials taken into account their body and surface [87]. Furthermore, absorption and adhesion properties, as well as the hydrophobicity, etc. features should be taken into account [88]. Formation of the novel direction to study the structured materials used in the optoelectronics devices are important as well [89]. The relationship of the properties of new materials with their usefulness for the vital activity of a living organism is important when searching for new ways to study of the nanomaterials [90].

## 4. Conclusions

To summarize the obtained results, it should be concluded that:

The process of sensitization significantly affects the change in the refractive parameters of the organic conjugated materials with the initially existing donor–acceptor interaction. The refractive parameters can be changed up to some order of magnitude. This can be explained by the formation of the additional dipole moment, which influences the basic physical–chemical parameters of the doped materials. This dipole moment correlates with the increased transferred charge and increased the barrier-free pathway for the electrons via intermolecular CTC formation (µ = *q* × *L*).

The process of sensitization can predict the following consideration. It is possible to say that the elementary materials volume associated with the lattice parameters of the inorganic materials may correlate well with the local volume of the medium obtained by the irradiation of the organic matrices with the introduced nanostructures. The interference pattern from two beams, recording the gratings, provides the diffraction due to the incorporated nanostructures, creating a 3D environment. It is not real crystal axes, but it can be used in the consideration.

The process of sensitization dramatically changes the surface relief of the solid organic films. It can be checked via the measurement of the wetting angle of the water drops at the organic films surface. The larger content of the nanoparticles incorporated in the materials body provokes the larger wetting angle. It permits us to find the tendency to increase in the hydrohobility of the surfaces and observe the “lotus effect”. Thus, it can be considered as the possible variant to protect different materials from corrosion. Indeed, with the dense packing of the nanoparticles inside the matrix, the skeleton of, say, the carbon nanotubes form a palisade (forest) of the protrusions on the surface, which will not allow the water droplets to penetrate into the material.

Moreover, it can be used in the explanation of the technical process connected with the nonlinear absorption. It can be considered as an additional optical limiting mechanism to protect the human eyes and technical devices from high laser irradiation in the complex laser schemes due to some energy losses via reflection from the structured surfaces.

It is important to note that the model matrix system with an initial donor–acceptor interaction requires the doping of the matrix at different concentrations to observe the effect of refractive parameters and wetting-angle changes. Some parts of the sensitizers will be included in the intermolecular charge transfer process, but the extra nano-objects, which are not involved in the process of complex formation, will be involved in the surface modification procedure. The last one permits us to use the structured polymers as separate elements in complex optoelectronic schemes. Moreover, the relief obtained can be used in the display technique to orient the liquid crystals without any additional alignment layers.

The nanotechnology approach shown can extend the area of the application of the studied materials, which can be useful for general optoelectronics and biomedicine. From the last point of view, this can be considered as a system of protruding pentagonal and hexagonal rings from the structured surface of doped plastic that does not allow viruses or bacteria with sizes from 5 to 200 nm to penetrate into such nanosurfaces.

The nonlinearities and charge carrier mobility parameters estimated for the studied organic materials can be compared with similar nonlinear and conductive parameters for the inorganic Si-based and LiNbO_3_ systems. It provokes a concurrent role and better design of the thin-film plastic optoelectronic elements than the volumetric inorganic construction.

## Figures and Tables

**Figure 1 polymers-15-02819-f001:**
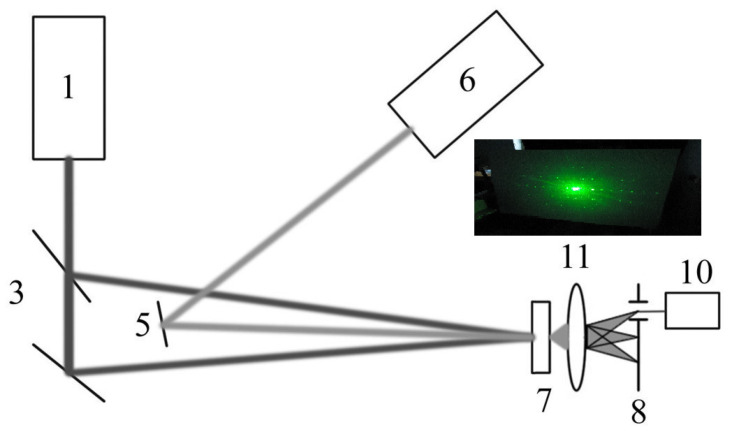
Holographic set-up to study the photorefractive parameters of the materials.

**Figure 2 polymers-15-02819-f002:**
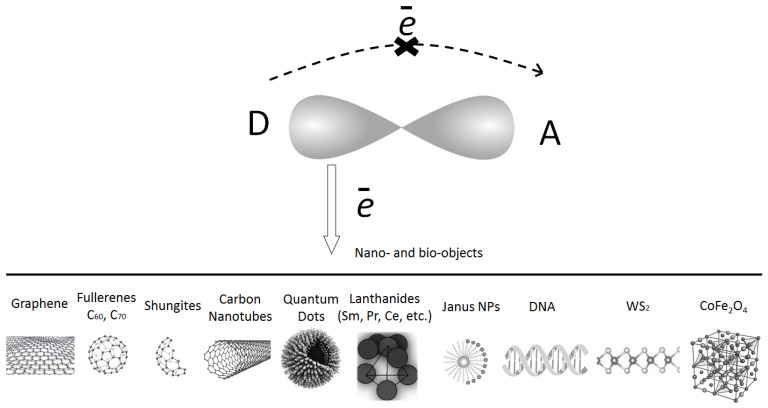
Qualitative view of possible CTC formation for the systems with the initial D–A *intera*ction (the **upper** part of figure); complicated charge transfer process with the formation of the *inter*molecular interaction (the **lower** part of figure).

**Table 1 polymers-15-02819-t001:** The division of the materials by their photoconductive properties.

Type of Materials	Conductivity, Ohm^−1^ × cm^−1^	Content, cm^−3^	Charge Carrier Mobility, cm^2^ × V^−1^ × s^−1^
Metals	10^2^…10^8^	10^22^	10^3^
Inorganic semiconductors	10^3^…10^−9^	10^11^…10^20^	10^5^…10^−3^
Organic semiconductors	10^2^…10^−14^	10^6^…10^19^	10^2^…10^−6^
Dielectrics	Less than 10^−14^	Less than 10^9^	Less than 10^−4^

**Table 2 polymers-15-02819-t002:** Comparative data of laser-induced change of the refractive index Δ*n_i_* for the currently studied and previously investigated plastic materials at the wavelength of 532 nm.

Studied Structure	Dopant Content wt.%	Energy Density, J × cm^−2^	Λ, mm^−1^	Laser Pulse Width, ns	Δ*n_i_*	References
Pure PI	0	0.6	90	20	10^−4^–10^−5^	[49]
PI + C_60_	0.2	0.5–0.6	90	10–20	4.2 × 10^−3^	[49]
PI + C_70_	0.2	0.6	90	10–20	4.68 × 10^−3^	[49]
PI + Sh ^1^	0.1	0.5	100	20	4.1 × 10^−3^	current
PI + Sh	0.2	0.5	100	20	5.5 × 10^−3^	current
PI + Sh	0.2	0.6	150	20	5.3 × 10^−3^	[59,60]
PI + CNTs	0.1	0.5–0.8	90	10–20	5.7 × 10^−3^	[49]
PI + CNTs	0.2	0.6	100	20	5.5 × 10^−3^	current
PI + DWCNTs ^2^	0.1	0.1	100	10	9.4 × 10^−3^	[61]
PI + DWCNTs	0.2	0.3	100	10	9.8 × 10^−3^	current
PI + DWCNTs	0.1	0.1	150	10	7.0 × 10^−3^	[61]
PI + RGO	0.1	0.1	100	20	7.5 × 10^−3^	current
PI + RGO	0.2	0.3	100	20	7.7 × 10^−3^	current
PI + RGO	0.1	0.1	150	20	6.5 × 10^−3^	current
Pure COANP	0	0.9	100	20	~10^−5^	[49,62]
COANP + C_60_	5	0.9	90–100	20	6.2 × 10^−3^	[62]
COANP + C_70_	0.5	0.9	100	20	4.5 × 10^−3^	current
COANP + C_70_	2.0	0.9	100	20	5.2 × 10^−3^	current
COANP + C_70_	5	0.9	90	10–20	6.89 × 10^−3^	[49]

Sh ^1^—shungite nano-object dopant, DWCNTs ^2^—double-walled carbon nanotubes dopant.

**Table 3 polymers-15-02819-t003:** Wetting angle change at the different organic thin films doped with the nano-objects.

Materials	Content of Sensitizers, wt.%	Sensitizers Type	Wetting Angle before Sensitization	Wetting Angle after Sensitization	References
PI	0.2	C_70_	72–73	84–85	current
PI	0.5	C_70_	72	89–90	[72]
PI	0.5	C_60_	72–73	89	current
PI	0.1	CNTs	75–79	101	[76]
COANP	0.5	C_70_	78–79	85–86	current
COANP	2.0	C_70_	78	92–93	current
PMPS ^1^	0.83	C_60_	75	81	[74]
PVA ^2^	0.1	C_60_	40	83	[74]
PVA	1.0	CNTs	39–40	82	[74]
NPP ^3^	1.0	C_60_	97	102	[74]
PNP ^4^	1.0	C_70_	90–91	94	[74]

PMPS ^1^—poly(methyl phenyl silane); PVA ^2^—polyvinyl alcohol; NPP ^3^—*N*-(4-nitrophenyl)-(L)-prolinol; PNP ^4^—2-(n-prolinol)-5-nitropyridine.

## Data Availability

Publicly available data on laser holographic technique can be found in https://doi.org/10.1023/A:1018506528934; some refractive parameters studied with different techniques correlated to each other can be seen in https://doi.org/10.1134/1.1509823.

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
