# Peer review of "Refractive Properties of Conjugated Organic Materials Doped with Fullerenes and Other Carbon-Based Nano-Objects"

_polymers, 2023, doi:10.3390/polym15132819_

Round 1

Reviewer 1 Report

In this manuscript, Natalia et al. reported some nanotechnology approach to increase the refractive and photoconductive parameters of the organic conjugated materials and the sensitization process and the laser treatment to improve the basic physical-chemical properties of the laser, solar energy, and general photonics materials. The large refractive index change in the doped organic conjugated systems are discussed. The comparative results of the refractivity and charge carrier mobility change are investigated for the doped polyimides and pyridine materials sensitized with the fullerenes, and reduced graphene oxides. The work is interesting for this field. I am pleased to recommend this manuscript acceptable for publication in Polymers after well addressing the following issues:

1. The author claimed that the role of the fullerene C60 in controlling the performance of inverted perovskite solar cells was shown in paper [21]. The fullerene derivatives of PCBM[60] or PCBM[71] was commonly used in photomultiplication type photodetectors, the related works should be cited, such as 10.1002/adfm.202212149; 10.1016/j.cej.2022.134973; and 10.1021/acsami.2c12154.

2. The author claimed that “It can stimulate the efficient inter-molecular charge transfer complex (CTC) formation in the nanoobjects-doped organic conjugated materials.” How to confirm the formation of CTCT state?

3. The intermolecular charge transfer complex formation process permits to find the drastic improvement of the spectral, photoconductive and dynamic parameters….. The more discussion on this phenomenon should be added or why is the drastic improvement?.

4. The more evidence should be provided to confirm that “considered as the possible variant to protect different materials from corrosion”.

More work should be carried out to improve English for better understanding. 

Author Response

Dear Reviewer!

Thank you very much for your recommendation. I am using them. Some papers are included in the text of the paper.

All corrected parts are collared with green.

Best Regards,

Natalia Kamanina

=======================================

Natalia V. Kamanina (Prof., Dr.Sci., PhD)

Head of the lab for Photophysics of media with nanoobjects

Vavilov State Optical Institute

Kadetskaya Liniya V.O., dom.5, korpus 2,

St.- Petersburg, 199053, Russia

Professor of the St.-Petersburg Electrotechnical University (“LETI”),

Part-time Leading Researcher at Nuclear Physics Institute (Gatchina)

Job phone: +7 (812) 327-00-95

Fax: +7 (812) 331-75-58 (for N.V.Kamanina)

e-mail:  nvkamanina@mail.ru

Lab_cite: sites.google.com/view/photophysics-lab

https://publons.com/researcher/1696479/natalia-kamanina/

https://sciprofiles.com/news-feed

http://rusnor.org/network/webinars/10203.htm

http://www.npkgoi.ru/?module=articles&c=profil&b=7

http://www.nanometer.ru/2007/08/09/liquid_crystal_3905.html

http://www.eltech.ru/ru/fakultety/fakultet-elektroniki/sostav-fakulteta/kafedra-kvantovoy-elektroniki-i-optiko-elektronnyh-priborov/sostav-kafedry

=======================================

Comments and Suggestions for Authors

In this manuscript, Natalia et al. reported some nanotechnology approach to increase the refractive and photoconductive parameters of the organic conjugated materials and the sensitization process and the laser treatment to improve the basic physical-chemical properties of the laser, solar energy, and general photonics materials. The large refractive index change in the doped organic conjugated systems are discussed. The comparative results of the refractivity and charge carrier mobility change are investigated for the doped polyimides and pyridine materials sensitized with the fullerenes, and reduced graphene oxides. The work is interesting for this field. I am pleased to recommend this manuscript acceptable for publication in Polymers after well addressing the following issues:

  1. The author claimed that the role of the fullerene C60 in controlling the performance of inverted perovskite solar cells was shown in paper [21]. The fullerene derivatives of PCBM[60] or PCBM[71] was commonly used in photomultiplication type photodetectors, the related works should be cited, such as 10.1002/adfm.202212149; 10.1016/j.cej.2022.134973; and 10.1021/acsami.2c12154.

These references have been included in the text body. They are now Refs [22-24].

  1. The author claimed that “It can stimulate the efficient inter-molecular charge transfer complex (CTC) formation in the nanoobjects-doped organic conjugated materials.” How to confirm the formation of CTCT state?

Yes, thank you! The paragraph has been included in the text body after the Fig.2. It should be mentioned, that for most of the organic structures doped with the indicated nanoparticles the additional experiments, such as mass-spectrometry analysis, color change testing, etc. have been made in order to support the intermolecular CTC formation.

An additional paragraph connected with the polyimide is included as well.

In addition, for example, the color of the polyimide varies from bright orange (pure structure) to bright brown (when sensitized with the fullerene C60) or to dark orange (when sensitized with the fullerene C70).

  1. The intermolecular charge transfer complex formation process permits to find the drastic improvement of the spectral, photoconductive and dynamic parameters….. The more discussion on this phenomenon should be added or why is the drastic improvement?.

Yes, I have added some discussion. The paragraph is included in the text body.

Really, once again, an increase in the barrier-free path of the charge carrier transfer from an intramolecular donor to an intermolecular acceptor leads to an increase in the mobility of charge carriers; this stimulates an increase in the photoconductivity. An increase in the dipole moment is accompanied by an increase in the absorption cross-section (the square of the dipole moment is proportional to the absorption cross-section); which is realized in a change in absorption and spectral features. Thus, there is a correlation between the photoconductive and refractive parameters of sensitized materials.

  1. The more evidence should be provided to confirm that “considered as the possible variant to protect different materials from corrosion”.

In conclusion part it has been added the paragraph:

Indeed, with the dense packing of the nanoparticles inside the matrix, the skeleton of, say, the carbon nanotubes forms a palisade (forest) of the protrusions on the surface, which will not allow the water droplets to penetrate into the material.

Thank you. It will be made in the future experiments. But now I have added some paragraph in the text body as well.

  1. Comments on the Quality of English Language

More work should be carried out to improve English for better understanding. 

Thank you! I have check English.

Reviewer 2 Report

I recommended publication without any changes 

Author Response

Dear Reviewer!

Thank you very much for your time, which you have spent for the estimation of the paper.

Thanks a lot for your kind reply

I have improved a little bit the paper.

All corrected parts are collared with green.

Best Regards,

Natalia Kamanina

=======================================

Natalia V. Kamanina (Prof., Dr.Sci., PhD)

Head of the lab for Photophysics of media with nanoobjects

Vavilov State Optical Institute

Kadetskaya Liniya V.O., dom.5, korpus 2,

St.- Petersburg, 199053, Russia

Professor of the St.-Petersburg Electrotechnical University (“LETI”),

Part-time Leading Researcher at Nuclear Physics Institute (Gatchina)

Job phone: +7 (812) 327-00-95

Fax: +7 (812) 331-75-58 (for N.V.Kamanina)

e-mail:  nvkamanina@mail.ru

Lab_cite: sites.google.com/view/photophysics-lab

https://publons.com/researcher/1696479/natalia-kamanina/

https://sciprofiles.com/news-feed

http://rusnor.org/network/webinars/10203.htm

http://www.npkgoi.ru/?module=articles&c=profil&b=7

http://www.nanometer.ru/2007/08/09/liquid_crystal_3905.html

http://www.eltech.ru/ru/fakultety/fakultet-elektroniki/sostav-fakulteta/kafedra-kvantovoy-elektroniki-i-optiko-elektronnyh-priborov/sostav-kafedry

=======================================

Open Review

Quality of English Language

(x) I am not qualified to assess the quality of English in this paper
( ) English very difficult to understand/incomprehensible
( ) Extensive editing of English language required
( ) Moderate editing of English language required
( ) Minor editing of English language required
( ) English language fine. No issues detected

Yes

Can be improved

Must be improved

Not applicable

Does the introduction provide sufficient background and include all relevant references?

(x)

( )

( )

( )

Are all the cited references relevant to the research?

(x)

( )

( )

( )

Is the research design appropriate?

(x)

( )

( )

( )

Are the methods adequately described?

(x)

( )

( )

( )

Are the results clearly presented?

(x)

( )

( )

( )

Are the conclusions supported by the results?

(x)

( )

( )

( )

Comments and Suggestions for Authors

I recommended publication without any changes 

Reviewer 3 Report

The author of this paper has conducted a study on the impact of doping carbon-based nanoparticles on the refractive and photoconductive properties of organic conjugated materials. This research is significant in the context of the growing demand for optoelectronic materials. The author has systematically examined the refractive characteristics and volt-current responses of the various doped materials. This study has the potential to advance nanotechnology-enabled photoconductivity improvements. Therefore, I recommend this paper for publication with only minor edits.

However, I have one concern regarding Figure 2. The author includes several nanostructures such as WS2 and CoFe2O4, which are not carbon-based nanomaterials. This inclusion goes beyond the scope of the manuscript, as the title indicates that the article focuses solely on carbon-based nanostructures. I suggest removing these non-relevant nanostructures from Figure 2 to ensure better alignment with the manuscript's objectives.

With this minor revision, the paper will be ready for publication.

Author Response

Dear Reviewer!

Thank you very much for your kind estimation of my paper.

 Sorry me please. I would like to save (keep) these nanoparticles (WS2 and CoFe2O4) in my model in order to have the chance to refer this paper as the background for the future investigations.

The paragraph has been added in the text body:

Of course, it is worth noting that nanoparticles such as WS2 and CoFe2O4 are not carbon-containing, but they are taken into account in this model for the possibility of referring to it in subsequent experiments.

I have improved it a little bit.

All corrected parts are collared with green.

Best Regards,

Natalia Kamanina

=======================================

Natalia V. Kamanina (Prof., Dr.Sci., PhD)

Head of the lab for Photophysics of media with nanoobjects

Vavilov State Optical Institute

Kadetskaya Liniya V.O., dom.5, korpus 2,

St.- Petersburg, 199053, Russia

Professor of the St.-Petersburg Electrotechnical University (“LETI”),

Part-time Leading Researcher at Nuclear Physics Institute (Gatchina)

Job phone: +7 (812) 327-00-95

Fax: +7 (812) 331-75-58 (for N.V.Kamanina)

e-mail:  nvkamanina@mail.ru

Lab_cite: sites.google.com/view/photophysics-lab

https://publons.com/researcher/1696479/natalia-kamanina/

https://sciprofiles.com/news-feed

http://rusnor.org/network/webinars/10203.htm

http://www.npkgoi.ru/?module=articles&c=profil&b=7

http://www.nanometer.ru/2007/08/09/liquid_crystal_3905.html

http://www.eltech.ru/ru/fakultety/fakultet-elektroniki/sostav-fakulteta/kafedra-kvantovoy-elektroniki-i-optiko-elektronnyh-priborov/sostav-kafedry

=======================================

Начало формы

Open Review

Quality of English Language

( ) I am not qualified to assess the quality of English in this paper
( ) English very difficult to understand/incomprehensible
( ) Extensive editing of English language required
( ) Moderate editing of English language required
( ) Minor editing of English language required
(x) English language fine. No issues detected

Yes

Can be improved

Must be improved

Not applicable

Does the introduction provide sufficient background and include all relevant references?

(x)

( )

( )

( )

Are all the cited references relevant to the research?

(x)

( )

( )

( )

Is the research design appropriate?

(x)

( )

( )

( )

Are the methods adequately described?

(x)

( )

( )

( )

Are the results clearly presented?

(x)

( )

( )

( )

Are the conclusions supported by the results?

(x)

( )

( )

( )

Comments and Suggestions for Authors

The author of this paper has conducted a study on the impact of doping carbon-based nanoparticles on the refractive and photoconductive properties of organic conjugated materials. This research is significant in the context of the growing demand for optoelectronic materials. The author has systematically examined the refractive characteristics and volt-current responses of the various doped materials. This study has the potential to advance nanotechnology-enabled photoconductivity improvements. Therefore, I recommend this paper for publication with only minor edits.

However, I have one concern regarding Figure 2. The author includes several nanostructures such as WS2 and CoFe2O4, which are not carbon-based nanomaterials. This inclusion goes beyond the scope of the manuscript, as the title indicates that the article focuses solely on carbon-based nanostructures. I suggest removing these non-relevant nanostructures from Figure 2 to ensure better alignment with the manuscript's objectives.

With this minor revision, the paper will be ready for publication.

 Sorry me please. I would like to save (keep) these nanoparticles in my model in order to have the chance to refer this paper as the background for the future investihgations.

Submission Date

05 June 2023

Date of this review

06 Jun 2023 05:23:17

Конец формы

© 1996-2023 MDPI (Basel, Switzerland) unless otherwise stated
